# Human Gaze is All You Need: Aligning Image Encoders with Human Attention

## Abstract

Replicating human-like perception in artificial systems requires capturing the attentional biases that shape human interpretation of visual scenes. While modern Vision-Language Models (VLMs) demonstrate strong multimodal reasoning, they often lack the behavioral priors that guide human attention. We address this gap with a framework that integrates human gaze patterns into the visual encoder of a state-of-the-art VLM. Aggregated attention heatmaps—collected from 29 participants in a visual description task—are incorporated via a cross-attention mechanism that refines the encoder's latent space to prioritize human-salient regions. Aligning model attention with human gaze yields consistent improvements in both human-likeness and semantic accuracy of image descriptions. **METEOR** and **Cosine Similarity** increase by *29.6%* and *4.6%*, respectively. Our contributions are threefold: a lightweight, plug-in **architectural modification of VLM** for integrating behavioral priors without full model retraining; empirical evidence of **enhanced alignment** with human perception, especially in scenes with strong bottom-up saliency cues; and a **novel dataset** of 778 image–heatmap–caption triples to facilitate research on attention-conditioned generation. This work demonstrates that incorporating behavioral priors systematically enhances VLMs and contributes to the development of more human-aligned interpretative capabilities for social cognition and human–AI interaction.

## 1 Introduction

Human perception is inherently selective. When viewing a scene, people allocate attention unevenly, focusing on the semantically or visually salient regions that guide their understanding and interpretation. This selectivity emerges from a complex interplay between bottom-up sensory cues and top-down cognitive goals, and plays a central role in tasks such as scene understanding and language generation Henderson & Hayes (2017). While Visual Language Models (VLMs) Li et al. (2023) have demonstrated impressive progress in multi-modal understanding, they typically lack access to behavioral signals that shape human interpretation. Consequently, their outputs often diverge from human-like descriptions, particularly in scenarios driven by stimulus-driven (bottom-up) attention.

In this work, we propose a novel architectural approach to bridge this gap by directly injecting human attention heatmaps into the visual encoder of a state-of-the-art VLM, Qwen2.5-VL Bai et al. (2025). Unlike previous methods that incorporate gaze data through auxiliary heads, additional supervision, or complex attention fusion modules, our method seamlessly integrates the behavioral signal into the latent space of the model through a calibrated injection point in the middle of the vision encoder. This preserves the pre-trained model architecture, avoids overhead in additional parameters, and allows the injected signal to interact natively with mid-level vision representations optimized for downstream generation.

To support this integration, we introduce a new dataset comprising 30 natural images sourced from CAT2000 Borji & Itti (2015), each paired with aggregated gaze heatmaps recorded from 29 human participants and human-generated image descriptions. This data set captures the diversity of human attention and provides a ground truth to evaluate the alignment between the model and human perception.

Our contributions are fourfold:

1. **Architectural modification**: We propose a lightweight yet effective injection mechanism for behavioral priors, enhancing VLMs without modifying their architecture or retraining from scratch (code repo available in the supplementary materials).

2. **Demonstrated behavioral alignment**: We show that injecting gaze heatmaps improves the human-likeness and semantic precision of generated captions, especially in scenes dominated by bottom-up saliency.

3. **Dataset release**: We release a unique dataset of image–heatmap–caption triples for studying attention-conditioned generation (dataset and annotations provided in the supplementary materials).

4. **Practical readiness**: Our system is optimized with FlashAttention for real-time applications such as virtual and augmented reality, enabling low-latency generation aligned with human focus.

Our experiments demonstrate that embedding human attentional priors directly into a VLM's encoder not only boosts standard captioning metrics, but also produces descriptions whose focus and intent more closely mirror how people perceive scenes. These results highlight the promise of behavioral-prior integration as a pathway toward more interpretable, contextually aware, and user-aligned vision–language systems.

## 2 RELATIONSHIP TO PRIOR WORK

Visual Question Answering (VQA) Antol et al. (2015); Yu et al. (2019) and Image Captioning Anderson et al. (2018); Sugano & Bulling (2016) are challenging multimodal tasks requiring a deep understanding of both visual content and associated text. Neural attention mechanisms Vaswani et al. (2017); Shih et al. (2016); Nam et al. (2017) are key for models to focus on relevant information. The field of human attention modeling has seen significant advances, with comprehensive surveys Cartella et al. (2024) documenting trends, applications, and challenges in integrating behavioral priors into computational systems.

In VQA, a crucial aspect has been the interplay between visual feature extraction and question understanding. Early approaches often relied on global image features. However, significant progress was made by incorporating more nuanced attention. **Bottom-up** attention mechanisms Anderson et al. (2018), for instance, first use object detectors (like Faster R-CNN) to identify salient regions in an image, effectively proposing a set of objects or areas of interest along with their features. These region-based features then serve as the basis for subsequent processing. Complementary to this, **top-down** attention mechanisms allow the question (the textual input) to guide the model's focus towards the most relevant of these proposed regions or features. The question semantics help to dynamically weight or select visual information pertinent to answering the query. This combination is powerful because the bottom-up process provides a rich, object-centric representation of the visual scene, while the top-down process ensures that the model selectively uses visual information that is most relevant to the specific linguistic query, mimicking how humans might first scan a scene for prominent objects and then focus their attention based on a specific question. Models like MCAN Yu et al. (2019) further refined this by using co-attention mechanisms to model complex interactions between question words and visual regions.

Despite the sophistication of these learned attention mechanisms, they do not always align with human gaze or reasoning patterns Chen et al. (2020); Das et al. (2016). This observation has motivated incorporating human-like attention as a supervisory signal or inductive bias, for example, by using predicted human saliency maps to guide network focus Selvaraju et al. (2019); Wu & Mooney (2019). Recent benchmarks like the AIM 2024 Challenge on Video Saliency Prediction Moskalenko et al. (2024) have advanced the state of saliency modeling, providing standardized evaluation protocols for attention prediction tasks.

When integrating human gaze heatmaps, VQA and especially Image Captioning Sugano & Bulling (2016) have proven to be highly indicative tasks. Image Captioning, in particular, allows models to generate a holistic description of an image's content and events, unconstrained by leading questions common in VQA. This setup offers a more realistic evaluation of a model's ability to ground language

in visual understanding, reflecting a more natural human-like interpretation. Our research, therefore, primarily focuses on leveraging human-like attention within the Image Captioning framework. Prior work, such as the Multimodal Human-like Attention Network (MULAN) Sood et al. (2021), has explored integrating human-like attention derived from models like TSM Sood et al. (2020) for text and MDS Fosco et al. (2020) for images, for both text and images in VQA by modifying attention scores within self-attention layers of an MCAN-based Yu et al. (2019) architecture.

Recent approaches have explored alternative strategies for attention integration. GazeLLM Rekimoto (2025) simulates human focus through object detection and image cropping without requiring external gaze data, while Voila-A Yan et al. (2025) uses mouse-tracking as a gaze proxy within BLIP-2/OpenFlamingo architectures that repeatedly fuse visual and linguistic modalities. Other work like "MLLMs Know Where to Look" Zhang et al. (2025) focuses on training-free perception of fine-grained details. While these approaches share the broader goal of human-aligned vision-language modeling, they differ fundamentally from our methodology in both architectural choices and problem formulation.

Our work diverges by proposing a more direct integration of contextual information (derived from or guided by human attention insights for captioning) into the visual processing pipeline. We leverage **cross-attention**, as originally formulated in the Transformer architecture Vaswani et al. (2017), to inject this information into the visual encoder. Here, visual features are Keys ($K_{vis}$) and Values ($V_{vis}$), and contextual information (e.g., initial caption context or high-level human gaze-derived concepts $E_{ctx}$) is the Query ($Q_{ctx}$). The output, contextually modulated visual features, is then integrated into the visual encoder. This method avoids extensive patching of self-attention layers and offers a dedicated fusion mechanism.

This approach is also motivated by insights from mechanistic interpretability, where specific attention heads and MLP layers in Transformers can form "circuits" for distinct sub-tasks Elhage et al. (2021). By using cross-attention, we aim to guide these visual processing circuits, making them more attuned to caption-relevant visual patterns from an early stage.

This architectural choice is informed by recent discussions on the role of attention in both artificial and biological systems. The Transformer architecture replaced recurrence and convolution with multi-head self-attention, enabling powerful modeling of long-range dependencies. This design has inspired comparisons with attentional processes in humans. For example, large-scale studies like those by Lai et al. Lai et al. (2019) have shown partial overlap between human gaze and machine attention maps, with results suggesting that incorporating human attention can improve performance and robustness, especially in attention-driven tasks.

In cognitive neuroscience, attention is known to influence both neural dynamics, such as gamma band oscillations in primate area V4 that predict response speed, and behavioral outcomes in humans Fries et al. (2001). In psycholinguistics, attention helps structure how language is processed and produced, with syntactic relationships guiding stimulus-driven attention Yuan et al. (2024). Despite these insights, there is still little understanding of how manipulating a model's attention maps might causally influence its linguistic behavior. Most of the current work focuses on observing attention rather than using it as a tool for targeted intervention.

To move beyond this, we treat attention not just as a learned weighting, but as a functional and editable data structure. Rather than simply reweighting inputs, we integrate aggregated human attention maps directly into the model's latent space through cross-attention. This allows us to align model representations with human attentional patterns at both the pixel and feature levels. Importantly, we observe systematic changes in the model's language output: the generated text becomes more aligned with human interpretive strategies. By integrating gaze-inspired priors early in the visual encoder, our method offers a biologically motivated and mechanistically interpretable route toward aligning model perception and generation with human attentional and interpretive norms.

## 3 METHODS

### 3.1 DATASET DEVELOPMENT: HUMAN-ALIGNED VISUAL DESCRIPTIONS

To ground our model's visual attention in real human behavior, we created a novel dataset comprising 30 unique images from the CAT2000 dataset Borji & Itti (2015), spanning five distinct visual

categories: **Abstractions** (Patterns)**, Social, Object, Compositions,** and **Disrupted** (Low Resolution). These categories were chosen to activate diverse attentional systems: Abstractions and Disrupted primarily engage bottom-up, stimulus-driven attention, while Social, Object, and Compositions engage top-down, goal-directed attention. For each image, we collected synchronized human eye-tracking data and verbal image descriptions from 29 participants during a structured two-phase viewing and recall task. Participants first freely observed an image, then described it aloud while being recorded.

These categories also differ in object numerosity:

- **Zero-object scenes:** Abstractions and Disrupted often lack identifiable objects.
- **One-object scenes:** Object images typically contain a single dominant item.
- **Multi-object scenes:** Social and Compositions involve multiple salient entities.

This dual categorization enables analysis of how attentional mechanisms and scene complexity jointly shape visual saliency.

This setup allowed us to extract heatmaps of aggregated human gaze patterns aligned with natural language outputs. Each session was recorded using high-fidelity eye-tracking hardware and resulted in verbal and visual behavioral traces, yielding 778 high-quality (image, heatmap, caption) triplets after filtering. The dataset serves as the foundation for training and evaluating our attention-injection framework.

Full details of the recording protocol, hardware, pre-processing, and participant instructions are provided in the Appendix.

### 3.2 ARCHITECTURE MODIFICATION FOR SALIENCY INTEGRATION

Our approach modifies a Vision-Language Model (VLM), exemplified by architectures like Qwen-VL, InternVL Bai et al. (2025); Chen et al. (2024), to incorporate human saliency information ($M_{\text{sal}} \in \mathbb{R}^{H_{\text{img}} \times W_{\text{img}} \times 1}$). VLMs typically process an image $I_{\text{img}}$ through a Vision Encoder (VE) to produce visual embeddings $Z_v \in \mathbb{R}^{N_v \times d_{\text{model}}}$, which are then injected into a prompt template (e.g., $S_{\text{prompt}} = $ "Image: $\langle P_1 \rangle \dots \langle P_{N_v} \rangle$ Question: ... Answer:") at predefined placeholder locations for processing by the Large Language Model (LLM).

**Limitations of Alternative Injection Strategies.** Common strategies for incorporating auxiliary visual information, such as saliency maps, include concatenating them to the input image channels or directly modifying the self-attention scores within the VE (e.g., as in MULAN). However, these methods can lead to challenges such as pre-training mismatch with VEs typically expecting 3-channel inputs, increased computational overhead, or direct alteration of the VE's core self-attention mechanisms Dosovitskiy et al. (2021). A more detailed discussion of these alternatives, including the standard self-attention mechanism, is provided in the Appendix.

This architectural choice is inspired by insights from classical transformer architecture Vaswani et al. (2017), mechanistic interpretability Elhage et al. (2021). We inject the saliency signal (a "human ViT" representation) via this cross-attention "calibrator" within a VE layer. Integrating it near the VE's output, after several layers of self-attention have processed the image and a polysemantic space of high-order visual features has already formed, allows the saliency to refine these existing complex representations. This strategy aims to efficiently guide the VE's internal visual processing "circuits" towards human-perceived regions of interest with minimal disruption. The rationale for patching a specific number of layers and the choice of integration point are discussed in the Appendix.

**Proposed Saliency Injection via a Patched Transformer Layer with Cross-Attention.** We modify existing Vision Encoder (VE) layers by inserting a cross-attention mechanism after its self-attention sub-layer and before its feed-forward network (MLP) sub-layer. This "patched layer" design allows the saliency information to act as a calibrator on the features processed by self-attention, without fundamentally altering the nature of the feature vectors passing through the original layer's components.

Let $N_s = N_p$ is number of saliency patches and $d_{vit}$ is dimenition of VE's latent space. Thus, features $M'_{\text{sal}} \in \mathbb{R}^{N_s \times d_{\text{vit}}}$, derived from the input saliency map $M_{\text{sal}}$ via a learnable projection (as described

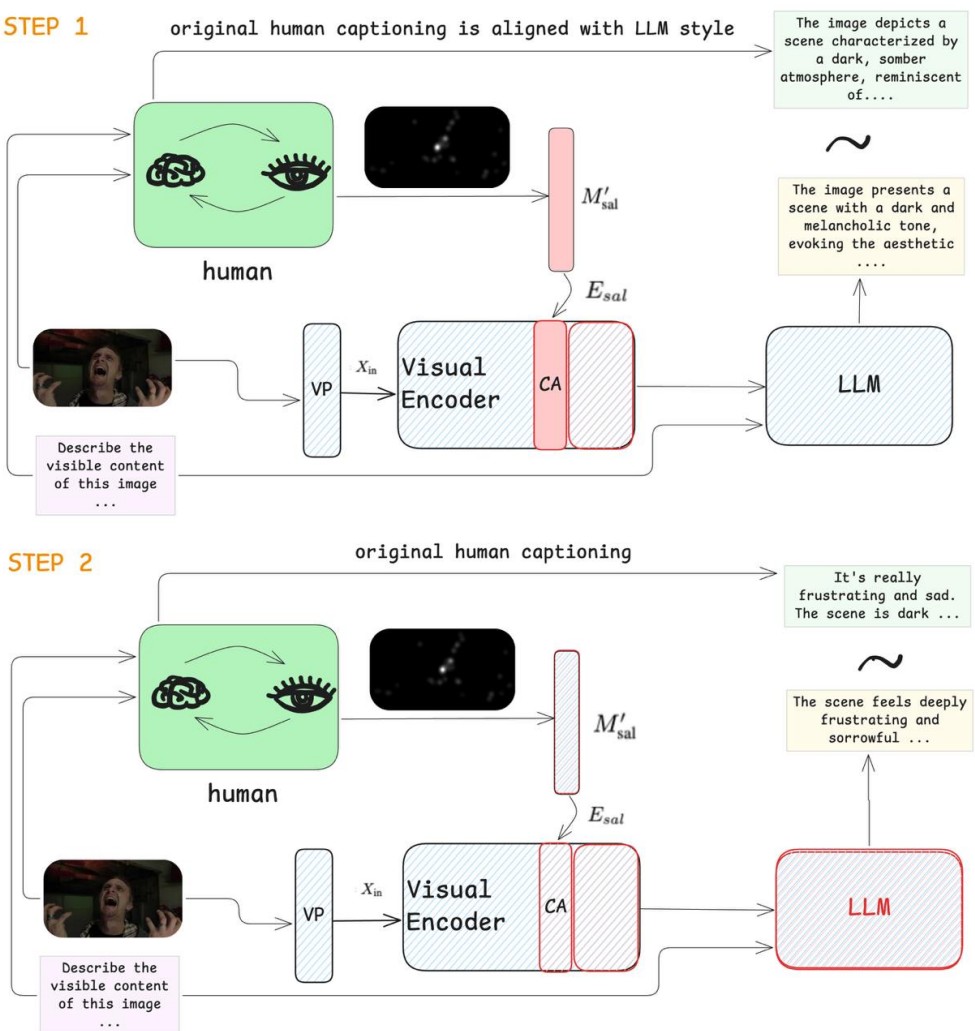

Figure 1: Two-stage training for saliency integration. **Step 1 (Saliency Calibrator Warm-up):** Human saliency features $M'_{\text{sal}}$ (derived via learnable $E_{\text{sal}}$) are injected into the Visual Encoder (VE) using an aditional Transformer Layer with Cross-Attention (CA) module. $E_{\text{sal}}$ and CA are trained from scratch (solid red); subsequent VE layers are LoRA fine-tuned (hatched red). The LLM (blue) is frozen; training uses LLM-stylized human captions. **Step 2 (Full Model Style Transfer):** Original human captions are used. Previously tuned components ($E_{\text{sal}}$, CA, VE layers via LoRA) and the LLM (now also LoRA fine-tuned, hatched red) are trained to adapt to human linguistic style and the integrated saliency. Color legend: Solid Red = trained from scratch; Hatched Red = LoRA fine-tuning; Blue = frozen.

by $E_{\text{sal}}$, detailed in the Appendix), serve as the Query for this inserted cross-attention mechanism. The output of the preceding self-attention sub-layer within the patched VE layer, $X'_{\text{self-attn}} \in \mathbb{R}^{N_p \times d_{\text{vit}}}$, acts as Keys and Values. The cross-attention mechanism then produces saliency-informed features:

$$Z'_{\text{sal\_informed}} = \text{CrossAttn}(Q = M'_{\text{sal}}, K = X'_{\text{self-attn}}, V = X'_{\text{self-attn}}) \in \mathbb{R}^{N_s \times d_{\text{vit}}}$$

These features $Z'_{\text{sal\_informed}}$ are then integrated with $X'_{\text{self-attn}}$ (e.g., via addition and layer normalization) before being passed to the MLP sub-layer of the patched Transformer layer. The full mathematical formulation of the multi-head cross-attention mechanism itself is standard Vaswani et al. (2017) and detailed in the Appendix.

**Training Strategy: Two-Stage Calibration and Style Transfer.** Our model is trained in a two-stage process (illustrated in Figure 1) to first calibrate the VE to human saliency and then adapt the entire VLM to human linguistic nuances.

STAGE 1: SALIENCY CALIBRATOR WARM-UP. The goal is to integrate $M_{sal}$ effectively, minimizing influence from human linguistic style variations. The learnable saliency projection (involving $E_{sal}$), the inserted cross-attention mechanism within the patched VE layer(s), and the LoRA Hu et al. (2021) adapters for these patched layers (including their MLP sub-layers) and any subsequent VE layers are trained. The LLM remains frozen. Training targets are human-provided image descriptions stylistically normalized by the pre-trained VLM.

STAGE 2: FULL MODEL STYLE TRANSFER AND REFINEMENT. This stage adapts the entire VLM to original human-generated captions. The components trained in Stage 1 continue to be LoRA-tuned. Additionally, selected layers of the LLM are now also fine-tuned using LoRA. This allows the model to better utilize the saliency-informed visual features and adapt its language generation to human stylistic nuances. This two-stage approach encourages the model to genuinely react to the injected saliency signal rather than merely memorizing response patterns.

# 4 RESULTS

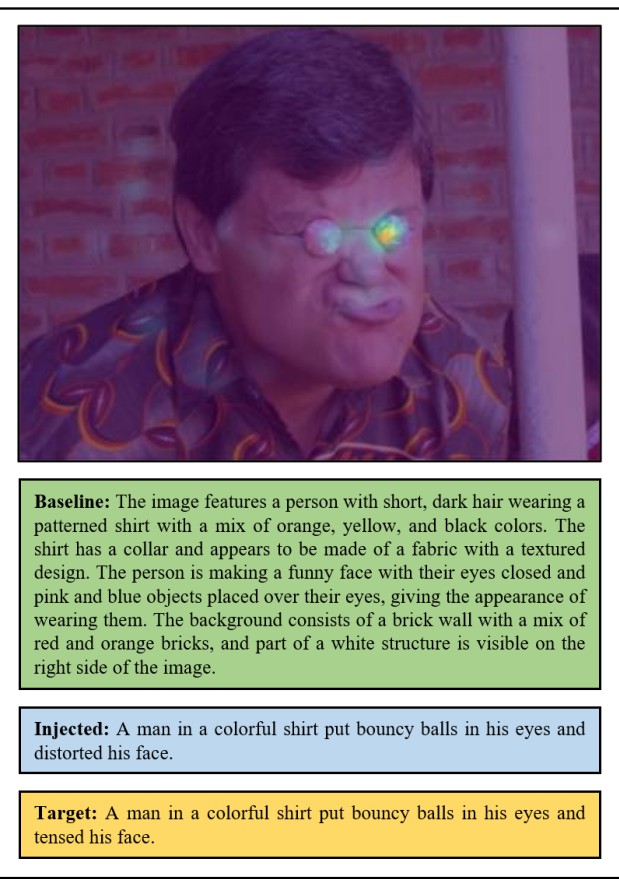

Figure 2: This image demonstrates the effect of human attention alignment. The **baseline** (an answer from VLM without injected saliency map) description is compared with **injected** (an answer from VLM with saliency map) and **target** (human answer) descriptions.

Our integration of human attention heatmaps into the VLM's visual encoder yields marked improvements across both aggregate and category-specific metrics. Table 1 shows the overall metric

comparison, with notable increases in ROUGE-L (+161.5%) and ROUGE-2 (+391.4%). Meanwhile, the +4.6% boost in Cosine Similarity reflects better semantic alignment with human-written captions.

Table 1: Metric Comparison for Overall Category

|  | METEOR | ROUGE-L | ROUGE-2 | ROUGE-1 | CS-F1 | CS-Recall | CS-Precision |
|---|---|---|---|---|---|---|---|
| Baseline | 0.12 | 0.11 | 0.02 | 0.13 | 0.85 | 0.88 | 0.82 |
| Injected | **0.15** | **0.29** | **0.15** | **0.32** | **0.89** | **0.90** | **0.89** |
| Diff (%) | +29.6% | +161.5% | +391.4% | +137.0% | +4.6% | +1.5% | +7.6% |

Table 2 highlights category-specific improvements, with the Abstraction category showing the highest METEOR increase (+55.9%) and Composition the highest CS-F1 gain (+5.9%).

Table 2: Metric Improvements by Category (%, Higher is Better)

| Category | METEOR | ROUGE-L | ROUGE-2 | ROUGE-1 | CS-F1 | CS-Recall | CS-Precision |
|---|---|---|---|---|---|---|---|
| Abstraction | +55.9% | +256.6% | +1583.3% | +207.0% | +4.8% | +2.0% | +7.5% |
| Object | -9.5% | +87.4% | +102.2% | +80.8% | +3.2% | -0.3% | +6.7% |
| Social | +13.4% | +115.6% | +184.9% | +118.5% | +4.1% | +1.1% | +7.1% |
| Disrupted | +42.1% | +178.0% | +463.5% | +127.8% | +4.8% | +2.1% | +7.5% |
| Composition | +51.9% | +204.1% | +522.7% | +172.9% | +5.9% | +2.5% | +9.4% |

\* CS — Cosine Similarity

Figure 2 illustrates the effect of this alignment with an example image and its corresponding descriptions.

ABLATION STUDY. To isolate and validate the impact of our core contribution—the saliency integration mechanism—we conducted an ablation study. This study is inherent to our experimental design, directly comparing the performance of our full model ("Injected") against an architecturally identical baseline model where this mechanism was ablated (i.e., deactivated). A statistical analysis comparing the output distributions of the two models revealed a significant difference, yielding a **p-value of less than 1e-5**. This result confirms that the improvements detailed below are a direct and statistically significant consequence of integrating human attentional priors, rather than a product of chance.

## 5 DISCUSSION

### 5.1 ATTENTION AS A CAUSAL HANDLE ON LANGUAGE

Our results confirm that human gaze maps function as *editable inductive priors* within a transformer's mid-level representations. Prior work suggested that aligning model attention with gaze improves interpretability Lai et al. (2019). We extend these insights by directly *injecting* saliency features via cross-attention, leading to systematic shifts in caption semantics. This provides the first causal evidence that attention maps can steer *language* circuitry in vision–language transformers. This claim of causality is further substantiated by our ablation study, which, as shown in the results, confirmed the statistical significance of the module's impact ($p < 1e-5$)

### 5.2 EXPLAINING CATEGORY-SPECIFIC GAINS

We selected images and divided them into five categories: Abstractions and Disrupted (bottom-up), Social and Compositions (top-down), and single-Object scenes. Our dual taxonomy reveals:

- **Zero-object, bottom-up scenes** (e.g., patterns): Lacking semantic anchors, these benefit most from gaze injection, as saliency maps disambiguate where to focus.
- **Multi-object, top-down scenes**: Rich contextual cues reduce but do not eliminate gains, indicating partial overlap between human goals and learned priors.

- **Single-object scenes**: When a single dominant object anchors the description, gaze provides less additional signal.

## 5.3 METRIC SENSITIVITY AND SEMANTIC ALIGNMENT

While n-gram based metrics like ROUGE and METEOR demonstrate a marked increase in lexical overlap with reference captions, it is important to consider the nature of these improvements. We observed that the baseline model (without heatmap injection) often produced captions that differed considerably in length and stylistic expression compared to both human references and the captions generated by our attention-injected model. Such variations in output verbosity and style can dispro­portionately affect n-gram overlap scores, potentially leading to the large percentage gains seen in ROUGE metrics.

In this context, the consistent +4.6% improvement in Cosine Similarity (CS-F1) becomes particularly informative. As Cosine Similarity measures semantic relatedness by comparing vector embeddings of the generated and reference captions, it is less sensitive to exact word matches or structural differences and more indicative of true semantic alignment. Therefore, while the high gains in n-gram based metrics are noteworthy and align with metrics used in prior research, we posit that the improvement in Cosine Similarity provides a more robust and direct measure of the enhanced semantic quality and relevance of the captions generated by our attention-injected model. The increase in ROUGE and METEOR, while substantial, should thus be interpreted as complementary indicators, reflecting in part the model's ability to generate outputs that are not only semantically closer but also potentially more aligned in structure with human references, a change likely facilitated by the guidance from human attention.

## 5.4 BROADER BEHAVIORAL PRIORS

Beyond gaze, other neurobehavioral signals—EEG and fMRI—offer promising inductive priors. Recent EEG–fMRI synthesis work shows transformers can translate scalp signals into haemodynamic maps (Li et al., 2024), and self-attention CNNs have been adapted for EEG classification (Ma et al., 2024). Architecturally, these modalities can be embedded as query streams in cross-attention layers, aligning model representations with richer cognitive signatures.

## 5.5 LIMITATIONS

- **Dataset scale and diversity**: Our corpus, comprising 30 unique images and gaze data from 29 participants, offers a controlled benchmark for modeling core attention mechanisms. However, its limited scale constrains generalization across demographics, tasks, and device contexts. Crowdsourced eye-tracking platforms such as SALICON (Jiang et al., 2015) demonstrate the feasibility of scaling attention data collection. Future work should explore hybrid pipelines combining low-fidelity large-scale inputs with high-fidelity samples.

- **Domain transferability**: While our approach performs well on naturalistic visual stimuli, its efficacy in domain-specific contexts—such as radiology, satellite imagery, or histopathol­ogy—remains untested. These domains present distinct visual grammars, expert gaze patterns, and task structures, which may not align with attention priors learned from natural images. To ensure applicability in real-world decision-critical settings, future evaluations should incorporate task-aligned datasets with expert annotations.

- **Ethical and representational concerns**: Embedding human attention into AI models introduces a range of ethical considerations. Gaze data, being inherently biometric and behavioral, can be misused for surveillance or behavioral profiling (Liebling & Preibusch, 2014). Additionally, current eye-tracking corpora often underrepresent individuals with visual impairments, age-related changes, or neurodivergent attention profiles. This demo­graphic skew risks encoding and amplifying bias within attention-guided models (Chen et al., 2023). Mitigation strategies must include: (i) explicit informed consent and opt-in policies for gaze data collection; (ii) local, on-device inference pipelines to ensure user privacy; and (iii) fairness audits across subgroups to detect and correct distributional shifts.

## 5.6 FUTURE DIRECTIONS

Recent advances such as FlashAttention-2 (Dao, 2023) enable transformer inference below 50 ms on commodity GPUs, making real-time applications increasingly viable. We developed our solution directly with FlashAttention support and utilized a small and efficient base model such as Qwen-VL-3B-Instruct to open compelling opportunities for attention-guided systems in immersive and embodied contexts.

**Wearable and mobile deployments**: The integration of eye-tracking into head-mounted displays (HMDs)—exemplified by Apple Vision Pro, Meta Quest Pro, and Tobii-enabled AR glasses—offers unprecedented access to naturalistic, continuous gaze data in everyday settings. Such platforms can serve both as testbeds and deployment targets for real-time attention-aware AI, including mobile captioning, gaze-contingent rendering, and attention-modulated interaction design. Models trained on such data could optimize perceptual alignment between user intent and system response, enhancing usability and trust in ubiquitous computing.

**Assistive technologies**: In accessibility contexts, dynamic attention modeling holds promise for augmenting narration interfaces for visually impaired users. Systems could prioritize salient regions based on collective attention priors or individual user history, generating more informative scene descriptions. Further integration with speech synthesis and haptic feedback could enable multimodal assistive agents that adapt in real time to user attention.

**Medical and scientific imaging**: Attention-injected architectures offer potential in high-stakes domains like radiology and microscopy, where human gaze can serve as an interpretable prior for diagnostic relevance. Embedding gaze traces into training regimes or inference-time saliency modulation could improve both model performance and clinician trust, particularly when combined with explainable AI techniques.

**Multimodal behavioral priors**: Looking ahead, richer attention priors could emerge from aligning gaze with other biosignals—e.g., EEG, pupillometry, or even fMRI. This would enable attention modeling that reflects not only spatial selection but cognitive load, emotional state, and task engagement. Such multimodal inputs may inform more adaptive and human-aligned inductive biases in generative systems.

**Video and temporal dynamics**: The current focus on static image datasets leaves temporal attention modeling underexplored. Extending gaze-injection to video requires dynamic modeling of attention shifts, fixations, and re-entrance phenomena. Incorporating mechanisms for temporal continuity and memory may prove critical in developing attention-aware video captioning, scene understanding, or egocentric AI assistants.

In sum, our results suggest that human attention is not merely a useful input modality, but a manipulable inductive bias that can be explicitly injected into learning systems. By operationalizing this bias, we pave the way for AI that is more interpretable, context-aware, and grounded in human cognitive priors.

ETHICS STATEMENT

The collection and use of human gaze data in this research were conducted with rigorous adherence to ethical principles. The study protocol received approval from the institutional ethics committee, and all 29 participants provided written informed consent prior to their involvement. Participants were explicitly informed about the nature of the data being collected and its intended use for research purposes.

We acknowledge that the integration of human behavioral data, such as eye-tracking, into AI models carries significant ethical responsibilities. Gaze data is inherently biometric and can reveal sensitive behavioral patterns; therefore, its misuse for applications like surveillance or behavioral profiling is a valid concern. To mitigate this risk, all data was anonymized during preprocessing, and our research focuses exclusively on aggregated attention patterns rather than individual-level analysis.

Furthermore, we recognize that biases present in current eye-tracking corpora, which often underrepresent individuals with visual impairments, age-related visual changes, or neurodivergent attention profiles, can lead to the encoding and amplification of demographic biases within attention-guided models. While our dataset provides a controlled benchmark, its scale is limited. We advocate for future work to address these representational gaps, employing fairness audits and developing hybrid data collection pipelines to ensure more equitable and generalizable models. Our commitment is to foster the development of AI systems that are not only technologically advanced but also ethically robust and aligned with human-centric values.

REPRODUCIBILITY STATEMENT

To ensure full reproducibility of our findings, we have made all necessary components publicly available. The complete source code for our proposed architectural modification, including the cross-attention injection mechanism and the two-stage training strategy, is provided in the supplementary materials. This repository includes the necessary configuration files, Python scripts, and a Docker image to reconstruct the computational environment. To guide researchers and prevent potential ambiguity, a comprehensive README file is also included, detailing the setup process and execution steps. All experiments were performed on a single H100 GPU, and the provided assets allow for a direct replication of our model training, fine-tuning, and evaluation processes.

Furthermore, to allow for the replication of our human-aligned dataset, a detailed description of the data collection protocol is provided in Appendix A. This section outlines the participant recruitment criteria, the specific hardware used (including the EyeLink 1000+ eye-tracker), the structured two-phase experimental procedure for recording synchronized eye-tracking and verbal descriptions, and the data preprocessing steps. This comprehensive protocol provides a clear methodology for researchers to collect comparable gaze-caption datasets and offers full transparency on the generation of the 778 image-heatmap-caption triplets used in this work.

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

# A APPENDIX

## A.1 DATASET COLLECTION PROTOCOL

### A.1.1 PARTICIPANTS

We recruited 29 normotypic participants with normal or corrected-to-normal vision. Individuals who wore glasses or contact lenses were excluded to ensure high-fidelity eye-tracking. All participants provided written informed consent, and the study protocol was approved by the institutional ethics committee. Participants were compensated at a rate of $20 per hour for their time.

### A.1.2 EQUIPMENT

Stimuli were displayed on a 24-inch ASUS VG248QE monitor (1920 × 1080 resolution, 1 ms response time, 144 Hz refresh rate), with text rendered in 22-point Courier New font. Eye movements were recorded using the EyeLink 1000+ system (SR Research, 2024) at a sampling rate of 1000 Hz. Participants' heads were stabilized with a chin rest. The viewing distance was approximately 55 cm from the eye-tracking camera and 90 cm from the monitor, resulting in each character subtending a visual angle of approximately 0.29°. Only the dominant eye was tracked. Saccades and fixations were identified using SR Research's default saccade detection algorithm in the Data Viewer software. Stimulus presentation and data acquisition were controlled via SR Research Experiment Builder v2.1.140.

### A.1.3 PROCEDURE

Each session began with calibration procedures: chin rest positioning, dominant eye detection, focus adjustment, and a 9-point calibration sequence. Participants were instructed:

*"You will be shown 30 images, each presented twice. During the first presentation, look freely at the image for 5 seconds. Then, after a brief text prompt, the same image will reappear for 10 seconds. During this second presentation, please describe the image aloud as clearly and completely as possible."*

Drift correction was applied after each trial, where a trial consists of two consecutive presentations of the same image. Eye movements were recorded during all image presentations. Verbal descriptions were captured using a studio-grade microphone.

### A.1.4 EXPERIMENTAL DESIGN

Each participant completed 30 trials (5 categories × 6 images). The order of images was randomized per subject. The entire session lasted approximately 14 minutes. The design was within-subjects; all participants viewed the same images. Each trial produced both an eye-tracking recording and a synchronized verbal description.

### A.1.5 DATA PREPROCESSING

To ensure data quality, each sample (i.e., one image presentation) was manually reviewed. Exclusion criteria:

- Insufficient fixation density or technical artifacts in the heatmap.

- Inaudible or incomplete verbal descriptions.

After filtering, 778 valid samples were retained, each consisting of an image, aggregated human gaze heatmap, and aligned verbal description.

## A.2 ALTERNATIVE SALIENCY INJECTION STRATEGIES AND STANDARD ATTENTION

### A.2.1 LIMITATIONS OF INPUT CHANNEL CONCATENATION

One intuitive method to incorporate $M_{sal}$ is by concatenating it with the input image $I_{img}$ along the channel dimension, creating $I'_{img} \in \mathbb{R}^{H_{img} \times W_{img} \times (C_{img}+1)}$. However, this approach has significant drawbacks for modern pre-trained VLMs like Qwen-VL Bai et al. (2025) or InternVL Chen et al. (2024):

- **Pre-training Mismatch:** These VEs are typically pre-trained on images with $C_{img} = 3$ channels. Modifying the input channel count necessitates retraining the VE's initial layers.

- **Computational Cost and Parameter Increase:** Retraining or adapting the VE for a new input dimensionality increases parameters and computational expense.

### A.2.2 LIMITATIONS OF DIRECT SELF-ATTENTION SCORE MODIFICATION

Another class of methods involves directly modifying the attention scores within the VE's Vision Transformer (ViT) Dosovitskiy et al. (2021). Previous works, such as MULAN Sood et al. (2021), have incorporated human-like attention by directly modulating these attention scores. For an input token $i$ (corresponding to a query $q_i$) and its associated saliency-derived weight $\beta_i$, the scores are modified, for example:

$$\text{Scores}(q_i, K_j) = \frac{(q_i K_j^T) \cdot \beta_i}{\sqrt{d_h}}$$

While this integrates external guidance, it directly alters the self-attention's internal scoring mechanism.

### A.2.3 STANDARD MULTI-HEAD SELF-ATTENTION (MHA) IN VIT

Given an input sequence of patch embeddings $X_{\text{in}} \in \mathbb{R}^{N_p \times d_{\text{vit}}}$:

> Let $h$ be heads, $d_h = d_{\text{vit}}/h$.
>
> Projection matrices: $W_Q^j, W_K^j, W_V^j \in \mathbb{R}^{d_{\text{vit}} \times d_h}$ for head $j$.
>
> Output projection: $W_O \in \mathbb{R}^{d_{\text{vit}} \times d_{\text{vit}}}$.
>
> $Q_j = X_{\text{in}} W_Q^j, \quad K_j = X_{\text{in}} W_K^j, \quad V_j = X_{\text{in}} W_V^j$
>
> $A_j = \text{softmax}\left(\frac{Q_j K_j^\top}{\sqrt{d_h}}\right)$
>
> $\text{head}_j = A_j V_j$
>
> $\text{MHA}(X_{\text{in}}) = \text{Concat}(\text{head}_1, \ldots, \text{head}_h) W_O$

## A.3 SALIENCY PROCESSING AND PATCHED LAYER DETAILS

### A.3.1 SALIENCY FEATURE PROJECTION

The input saliency map $M_{\text{sal}} \in \mathbb{R}^{H_{\text{img}} \times W_{\text{img}} \times 1}$ is processed to derive features $M'_{\text{sal}} \in \mathbb{R}^{N_s \times d_{\text{vit}}}$ to be used as Queries for the cross-attention mechanism. This involves a learnable projection:

$$M'_{\text{sal}} = M_{\text{sal\_patches}} E_{\text{sal\_proj}} \in \mathbb{R}^{N_s \times d_{\text{vit}}} \tag{1}$$

where $M_{\text{sal\_patches}} \in \mathbb{R}^{N_s \times d_{\text{s\_raw}}}$ are embeddings derived from patches of $M_{\text{sal}}$, and $E_{\text{sal\_proj}} \in \mathbb{R}^{d_{\text{s\_raw}} \times d_{\text{vit}}}$ is the learnable projection matrix.

### A.3.2 PATCHED TRANSFORMER LAYER WITH CROSS-ATTENTION

A standard Transformer layer in the Vision Encoder (VE) consists of a Multi-Head Self-Attention (MHA) sub-layer followed by a Feed-Forward Network (FFN or MLP) sub-layer, with residual connections and layer normalization around each. Our modification, referred to as a "patched layer," inserts a Multi-Head Cross-Attention (MHCA) mechanism after the MHA sub-layer and before the FFN sub-layer.

Let $X_{\text{in\_layer}}$ be the input to the patched Transformer layer. 1. Self-Attention: $X'_{\text{self-attn}} = \text{MHA}(X_{\text{in\_layer}})$ (details of MHA in Appendix). $X_{\text{intermediate1}} = \text{LayerNorm}(X_{\text{in\_layer}} + X'_{\text{self-attn}})$. 2. Cross-Attention Injection (our patch): The Query for MHCA is $M'_{\text{sal}}$ (from Equation 1). The Key and Value for MHCA are $X_{\text{intermediate1}}$.

$$Z'_{\text{sal\_informed}} = \text{MHCA}(Q = M'_{\text{sal}}, K = X_{\text{intermediate1}}, V = X_{\text{intermediate1}})$$

The mathematical formulation of MHCA is analogous to MHA, with separate projection matrices for Q, K, V. Given $Q_{\text{src}} \in \mathbb{R}^{N_q \times d_{\text{vit}}}$ and $KV_{\text{src}} \in \mathbb{R}^{N_{kv} \times d_{\text{vit}}}$:

$$\text{Let } h_{\text{cross}} \text{ be heads, } d_{h,\text{cross}} = d_{\text{vit}}/h_{\text{cross}}.$$

$$W_Q^{\text{cross},k}, W_K^{\text{cross},k}, W_V^{\text{cross},k} \in \mathbb{R}^{d_{\text{vit}} \times d_{h,\text{cross}}}.$$

$$W_O^{\text{cross}} \in \mathbb{R}^{d_{\text{vit}} \times d_{\text{vit}}}.$$

$$Q_k = Q_{\text{src}} W_Q^{\text{cross},k}$$

$$K_k = KV_{\text{src}} W_K^{\text{cross},k}$$

$$V_k = KV_{\text{src}} W_V^{\text{cross},k}$$

$$A_k = \text{softmax}\left( \frac{Q_k K_k^\top}{\sqrt{d_{h,\text{cross}}}} \right)$$

$$\text{head}_k = A_k V_k$$

$$\text{MHCA}(Q_{\text{src}}, KV_{\text{src}}) = \text{Concat}(\text{head}_1, \ldots, \text{head}_{h_{\text{cross}}}) W_O^{\text{cross}}$$

In our case, $Q_{\text{src}} = M'_{\text{sal}}$ and $KV_{\text{src}} = X_{\text{intermediate1}}$. The output $Z'_{\text{sal\_informed}} \in \mathbb{R}^{N_s \times d_{\text{vit}}}$ is then integrated. If $N_s = N_p$ (e.g., saliency patches correspond to image patches), this integration can be: $X_{\text{intermediate2}} = \text{LayerNorm}(X_{\text{intermediate1}} + Z'_{\text{sal\_informed}})$. If $N_s = 1$ (global saliency query), $Z'_{\text{sal\_informed}}$ might be added (after replication) or used in a more complex fusion with $X_{\text{intermediate1}}$ to produce $X_{\text{intermediate2}}$. 3. Feed-Forward Network: $X'_{\text{ffn}} = \text{FFN}(X_{\text{intermediate2}})$. $X_{\text{out\_layer}} = \text{LayerNorm}(X_{\text{intermediate2}} + X'_{\text{ffn}})$.

### A.4 DETAILS OF THE TWO-STAGE TRAINING

#### A.4.1 STAGE 1: SALIENCY CALIBRATOR WARM-UP.

The primary goal of this stage is to enable the Vision Encoder (VE) to effectively integrate information from human saliency maps ($M_{\text{sal}}$) without being influenced by variations in human linguistic style. The raw saliency map $M_{\text{sal}} \in \mathbb{R}^{H_{\text{img}} \times W_{\text{img}} \times 1}$ is first processed into patch-like embeddings $M_{\text{sal\_patches}} \in \mathbb{R}^{N_s \times d_{\text{s\_raw}}}$. We then introduce a learnable embedding matrix $E_{\text{sal\_proj}} \in \mathbb{R}^{d_{\text{s\_raw}} \times d_{\text{vit}}}$ to project these into $M'_{\text{sal}} = M_{\text{sal\_patches}} E_{\text{sal\_proj}} \in \mathbb{R}^{N_s \times d_{\text{vit}}}$. Alternatively, if a single global saliency context vector is desired ($N_s = 1$), $M'_{\text{sal}} \in \mathbb{R}^{1 \times d_{\text{vit}}}$ is derived.

During this warm-up stage, the following components are trained:

- The saliency projection embedding $E_{\text{sal\_proj}}$ (or equivalent learnable parameters used to derive $M'_{\text{sal}}$).

- The entire cross-attention module (i.e., $W_Q^{\text{cross},j}, W_K^{\text{cross},j}, W_V^{\text{cross},j}, W_O^{\text{cross}}$).

- The layers of the VE that follow the injection point of $Z'_{\text{sal\_informed}}$, fine-tuned using LoRA Hu et al. (2021).

Crucially, for training targets in this stage, we use human-provided image descriptions that have been *stylistically normalized* by passing them through the pre-trained, frozen VLM to generate its own version of the captions. The LLM component of the VLM remains frozen.

#### A.4.2 STAGE 2: FULL MODEL STYLE TRANSFER AND REFINEMENT.

Once the saliency calibrator is warmed up, this stage adapts the entire VLM to human-generated captions. The components trained with LoRA in Stage 1 ($E_{\text{sal\_proj}}$, the cross-attention module, and subsequent VE layers) remain trainable with LoRA. Additionally, LoRA is applied to selected layers of the LLM.

The training data consists of original human-generated captions paired with images and saliency maps. By including the LLM in fine-tuning, the model learns to better utilize saliency-informed visual features and match human linguistic style.

## A.5 RATIONALE FOR SINGLE PATCHED LAYER INTEGRATION

The decision to integrate saliency information by patching a limited number of Vision Encoder (VE) layers, specifically favoring a single strategically placed patched layer, is informed by empirical observations and theoretical considerations related to model interpretability and optimization.

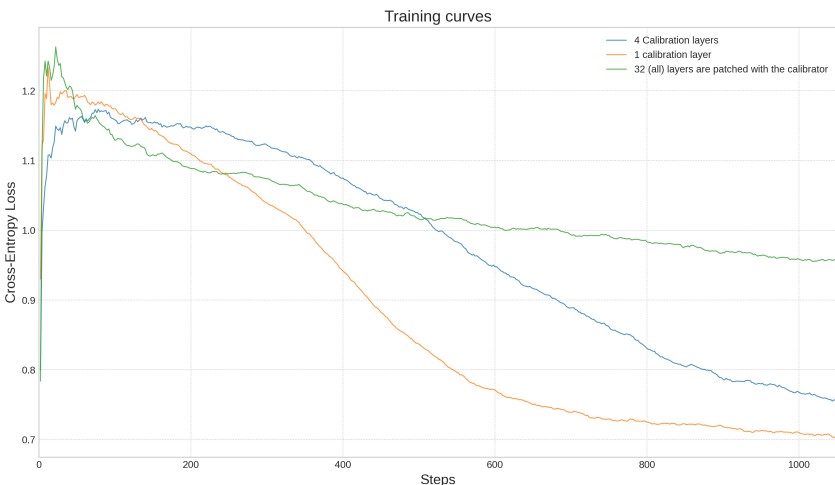

Figure 3: Loss curves during training for different numbers of VE layers patched with the saliency calibrator.

Figure 3 presents loss curves for three configurations:

- **32 Patched Layers (Green Line):** All available layers in the VE are patched with our cross-attention calibrator. This configuration demonstrates the highest initial loss and slower convergence, suggesting that modifying every layer introduces excessive complexity or instability, potentially disrupting the pre-trained features too extensively.

- **4 Additional Layers (Blue Line):** The cross-attention calibrator is inserted into four VE layers, distributed across the encoder. While performing better than patching all layers, it still shows a higher loss and less stable learning compared to a single patched layer.

- **1 Additional Patched Layer (Orange Line):** A single VE layer is patched. This configuration exhibits the lowest loss and the most stable and rapid convergence among the tested setups.

By this stage of the encoder, the model has likely formed high-level, polysemantic features, and the internal "circuits" Elhage et al. (2021) are processing more abstract visual concepts. Injecting the "human ViT" signal (our saliency-derived query) at this point allows the cross-attention mechanism to act as a fine-grained calibrator on these already sophisticated representations, rather than attempting to influence low-level feature extraction or overly perturbing the entire feature hierarchy. Modifying too many layers, especially early ones, might interfere with the foundational visual understanding learned during pre-training, leading to optimization difficulties as seen with the 32-layer and 4-layer patching strategies. A single, late-stage patched layer appears to strike an effective balance between incorporating external human-centric guidance and preserving the VE's powerful pre-trained capabilities.

All experiments were conducted on a single H100 GPU. To ensure robustness and reproducibility, the full experimental environment—including configuration files, Python scripts, and Docker images are available in the supplementary materials.

