# OpenReview forum: "Human Gaze is All You Need: Aligning Image Encoders with Human Attention"
_ICLR.cc/2026/Conference — Submitted to ICLR 2026_

### Official Review · Reviewer_DH94 · 2025-10-26

**Soundness:** 2
**Presentation:** 2
**Contribution:** 2
**Rating:** 2
**Confidence:** 4

**Summary:**

This paper proposed a method to align machine attention with human gaze in a VLM for image captioning task. The proposed method used a cross-attention module to integrate a saliency map into the vision encoder in a VLM. The experiments were performed on a dataset collected the authors. The results showed that Qwen2.5-VL with human gaze alignment performs better.

**Strengths:**

- The proposed an add-on architecture that preserves the VLM architecture.

**Weaknesses:**

- It is not clear why the authors collected a new dataset despite the existence of several other datasets [1], [2], [3].
- The evaluation is inadequate. The authors mentioned how previous works integrated human gaze. The proposed method should compare with them to show the proposed method is better than other human gaze integration techniques. The proposed method was compared with only one baseline. More baselines should be compared, this includes more VLMs and other image captioning methods that can incorporate gaze. Moreover, the evaluation is only performed on one dataset.

[1] Yang, Zheng, et al. "Eye-movement-prompted large image captioning model." Pattern Recognition 159 (2025): 111097.
[2] Vaidyanathan, Preethi, et al. "Snag: Spoken narratives and gaze dataset." Proceedings of the 56th Annual Meeting of the Association for Computational Linguistics (Volume 2: Short Papers). 2018.
[3] He, Sen, et al. "Human attention in image captioning: Dataset and analysis." Proceedings of the IEEE/CVF International Conference on Computer Vision. 2019.

**Questions:**

- More experiments on other datasets are needed.
- Where exactly is the heatmap integrated in the VE? It will be beneficial to have ablation studies on numbers of calibrators and location in the VE.
- Was the baseline finetuned on the collected dataset? If not, the results of a zero shot baseline and a finetuned baseline are needed to show the improvement comes from the human gaze alignment, not from finetuning a VLM.

---

### Official Review · Reviewer_Juda · 2025-10-30

**Soundness:** 1
**Presentation:** 3
**Contribution:** 2
**Rating:** 4
**Confidence:** 3

**Summary:**

This paper explores integrating human gaze heatmaps into a Vision–Language Model (Qwen2.5-VL) using a cross-attention module inside the visual encoder. It introduces a small dataset of 778 image-heatmap-caption triplets collected from 29 participants viewing 30 images. The method follows a two-stage training setup: (1) calibrating the visual encoder with saliency, and (2) adapting the model's language style to human captions. The authors report strong improvements on captioning metrics (e.g., +29.6% METEOR, +4.6% cosine similarity) and claim that gaze has a causal influence on language generation. The idea is interesting and nicely presented, but the dataset size and evaluation are too limited to support strong or general conclusions.

**Strengths:**

1) The idea is very interesting and relevant, and the line of research focused on improving VLMs through saliency and human attention is clearly at the state of the art. The fact that the authors use real gaze data and release it publicly is a strong plus. The motivation and positioning of the work are well justified and clearly introduced.
2) The paper is well written overall, with good structure and flow.
3) The code and dataset are publicly available, and the repository provided in the supplementary materials appears to be well documented. In addition, the paper includes a fair amount of methodological detail, which supports reproducibility.

**Weaknesses:**

# Related work (insufficient)

Regarding the related work, it would be clearer to have a dedicated section rather than embedding it in the introduction. Although lines 105-121 mention some of the most relevant works, it would strengthen the paper to also include other lines of research that aim to improve the alignment of VLLMs with humans, even if they do not explicitly use saliency. There is currently a growing body of work using DPO and RLHF for this purpose, for example:

* Ziyu Liu, Yuhang Zang, Xiaoyi Dong, Pan Zhang, Yuhang Cao, Haodong Duan, Conghui He, Yuanjun Xiong, Dahua Lin, and Jiaqi Wang. MIA-DPO: Multi-Image Augmented Direct Preference Optimization for Large Vision-Language Models. ICLR 2024. https://openreview.net/forum?id=f7WBRSuf9l

Lines 134-140 discuss the comparison between human and model attention, which is indeed relevant for this paper, but only one reference is provided. There are more recent works that analyze attention alignment in LLMs, such as Sood et al. and GLIMPSE:

* Guanxi Shen. GLIMPSE: Gradient-Layer Importance Mapping for Prompted Visual Saliency Explanation for Generative LVLMs. arXiv:2506.18985 (2025).
* Maxime Oquab et al. DINOv2: Learning Robust Visual Features without Supervision. TMLR (2023).

Additionally, there are other datasets on human attention that are not mentioned, for example:

* Ekta Sood et al. VQA-MHUG: A Gaze Dataset to Study Multimodal Neural Attention in Visual Question Answering. arXiv:2109.13116 (2021).

# Results and methodology concerns

1) The paper does not specify how the dataset of 778 image-heatmap-caption triples is divided for training, validation, and testing. It is unclear whether the split is by images, participants, or random sampling. Given that only 30 unique images are used, the lack of detail raises concerns about overfitting and the validity of the reported improvements.
2) The ablation study is described only briefly and lacks transparency. Although the authors claim statistical significance (p < 1e-5), they do not include any table, figure, or explanation of the statistical test used. Without quantitative details, the claim is not verifiable or reproducible.
3) From the description, it is unclear how the ablation experiment is performed, whether the model is trained identically but without the saliency injection module, or if the first calibration stage is omitted. This ambiguity makes it difficult to interpret what the reported comparison actually measures.
4) There is lack of comparison with recent baselines that also use gaze or saliency, such as Voila-A. It would be useful to evaluate whether the proposed method generalizes to their datasets or architectures, or to test it on multiple VLLMs.

# Limited experiments
1) Lines 508–522 describe the open-sourcing of code and data, which is excellent practice. However, this section could be shortened in the main paper to leave more room for experiments and analysis.

# Choice of dataset/task

The focus on image captioning is understandable, but the authors could briefly justify why this task was chosen instead of others like VQA, which is also highly relevant for gaze-based grounding.

# Synthetic human saliency

1) There is a large body of work on synthetic or generative saliency modeling, which is far more scalable than collecting real gaze data. For instance, Sood et al. use the MDSem model to generate synthetic saliency, and the VOILA-COCO dataset (also cited) is built automatically from Localized Narratives (LN-COCO) using GPT-4, making it much larger in scale than the dataset presented here.

2) A comparison with such synthetic saliency models or datasets would strengthen the paper. While collecting and releasing real gaze data is valuable, the dataset presented here is too narrow to generalize. One possibility would be to use it for fine-tuning an existing saliency model rather than treating it as a stand-alone benchmark.

# Minor comments
* Tables 1 and 2 should be adjusted to fit the page width.
* Avoid using two separate captions for the same table.
* Figures 1 and 2 could be made smaller to save space for more experimental content. Especially in Figure 1, the duplication of both stages could be avoided by indicating the two steps without repeating the entire diagram.

**Questions:**

* Could you clarify how the dataset of 778 samples is divided into training, validation, and test sets? Was the split done by images, by participants, or randomly? Given that there are only 30 unique images, how do you avoid overfitting? Did you perform any form of cross-validation or held-out evaluation?
* In the ablation study, what exactly changes between the baseline and the injected model? Is the training identical but without the saliency module, or is the first calibration step skipped?
* You mention a p < 1e−5 result, could you specify which statistical test was used and what was the sample size for that comparison? Where do you report these results?
* Have you compared your method with recent gaze- or saliency-based VLLMs such as Voila-A or GazeLLM? If not, could you discuss why or how your method might generalize to those settings?
* Why did you choose image captioning as the main task instead of others like VQA, which also strongly depend on attention and grounding?
* Have you considered testing your approach on synthetic or generated saliency maps (e.g., VOILA-COCO, MDSem) to evaluate whether it generalizes beyond real gaze data?

---

### Official Review · Reviewer_TjyZ · 2025-10-31

**Soundness:** 1
**Presentation:** 2
**Contribution:** 2
**Rating:** 2
**Confidence:** 4

**Summary:**

The paper proposes a method to align Vision-Language Models (VLMs) with human perceptual biases by integrating human gaze data directly into the visual encoder. The core contribution is an architectural modification: a lightweight, "plug-in" cross-attention module that is inserted into a single, pre-existing transformer layer within the vision encoder. This module acts as a "calibrator," using processed human gaze heatmaps as the Query and the vision encoder's intermediate features as the Key and Value to refine the model's representations.

**Strengths:**

- **Novel Architecture and training strategy**: The primary strength is the proposed architecture. Using a dedicated cross-attention mechanism as a "calibrator" inserted between existing self-attention and MLP blocks  is a clean, novel, and "plug-in" approach to integrating an external signal. And the two-stage training regimen is a good methodological contribution, correctly identifying the need to first align the vision encoder (Stage 1 ) before fine-tuning the language model for style (Stage 2).

**Weaknesses:**

- **Fatally Small Dataset:** The paper's claims are completely undermined by the dataset. 30 unique images  is not a sufficient basis for a general-purpose VLM alignment paper. It is a proof-of-concept on a toy problem. Any results from this are anecdotal and cannot be generalized. The model may have simply learned 30 "gaze-conditioned" caption styles.
- **Lack of Competitive Baselines:** The paper fails to compare its novel architecture against obvious and simpler alternatives. It only compares against "doing nothing". To prove the *method* is a contribution, it must be benchmarked against other ways to use gaze.
- **Misleading Paper Structure:** The paper's organization is poor. The most important experimental justification for the architecture (the 1-vs-4-vs-32 layer ablation in Fig 3 ) is relegated to the appendix.
- **Inflated Metrics:** The abstract and results heavily promote massive gains in ROUGE metrics (e.g., ROUGE-2 +391.4% ). However, the authors themselves admit in the discussion (Section 5.3) that these n-gram-based scores are likely inflated because the baseline model was stylistically different (e.g., more verbose). The metric that more robustly measures true semantic alignment, Cosine Similarity (CS-F1), shows only a modest **+4.6%** gain. This suggests the actual semantic improvement is far less dramatic than the ROUGE scores imply.

**Questions:**

See the weakness section, if the first and second issues could be addressed, I would consider to improve my socre

---

### Official Review · Reviewer_k3AG · 2025-11-01

**Soundness:** 2
**Presentation:** 3
**Contribution:** 2
**Rating:** 2
**Confidence:** 4

**Summary:**

This paper is motivated by the goal of aligning machine vision with human visual perception by explicitly modeling human gaze. To achieve this, the authors introduce a cross-attention mechanism that injects human saliency maps into the visual encoder of Qwen2.5-VL, allowing the model’s visual attention to be guided by human fixation patterns during caption generation. Experiments on a small self-collected gaze–caption dataset show that incorporating gaze information leads to noticeable improvements on standard captioning metrics.

**Strengths:**

1. The topic is timely and important: understanding and modeling human vision remains a central challenge in vision-language learning, and cognitive grounding via gaze is an underexplored yet meaningful direction.

2. The paper is clearly written and well-structured, making it easy to follow.

3. The authors show imaginative thinking in proposing future directions and potential extensions, such as integrating gaze-based attention alignment into multimodal interaction systems and exploring real-time, on-device applications for assistive or AR scenarios.

**Weaknesses:**

1. The experimental design introduces a shortcut by using human gaze maps as input while predicting captions that were also generated by those same humans. This setup inherently favors the model, since the gaze already encodes most of the caption content. Thus, the observed gains are unsurprising and do not convincingly show generalizable learning.

2. All results are based on one vision-language model (Qwen2.5-VL) and one custom dataset (778 samples from 30 images). No ablations are conducted on other models, datasets, or tasks, making it difficult to assess robustness or broader utility.

3. The paper claims to mimic human perception, but the model depends on externally provided gaze maps, effectively outsourcing perception rather than internalizing it. This setup enhances performance by using human input, rather than encouraging models to behave more like humans.

**Questions:**

1. The authors should clarify whether the method truly enables human-like perception, since it relies on explicit human gaze as input. It would be helpful to test with predicted saliency maps or without real gaze to see if the model still performs well.

2. All experiments are conducted only on Qwen2.5-VL, so it remains unclear whether the approach generalizes across different vision–language models. Testing on other backbones such as LLaVA or BLIP-2 would strengthen the claim of general applicability.

3. The evaluation focuses mainly on language metrics like METEOR and ROUGE, which do not directly measure human-likeness. The authors are encouraged to include perceptual or cognitive alignment metrics, or test on datasets such as AIR, to verify whether human-like attention improves reasoning rather than just caption overlap.

4. The experiments confirm that providing human gaze helps reproduce human captions, which is somewhat expected. The authors could add more analysis to show what deeper insights about visual attention or perception are gained from this work.

**Details Of Ethics Concerns:**

The authors have already addressed ethical considerations in the paper, noting that all participants provided written informed consent and that gaze data were anonymized. They also discussed potential privacy and representation issues and described reasonable safeguards, so no further ethical review appears necessary.

---

### Meta-Review · Area_Chair_xbzt · 2026-01-06

**Summary:**

The initial scores for this paper are all negative (2,2,2,4). The major concerns raised by the reviewers include: unsatisfying experiments (small data set of only 30 images, one vision-language model only, lack of baselines, no perceptual or cognitive alignment metrics), unconvincing motivation or problem setting (not surprising of improvement with human gaze as extra input, not using synthetic human saliency ), insufficient related works, etc.

**Reviewer Concerns:**

There is no author rebuttal.

**Reviewer Scores:**

There is no author rebuttal, so the scores are not expected to change.

---

### Decision · Program_Chairs · 2026-01-26

Reject